# Nearly-tight Approximation Guarantees for the Improving Multi-Armed Bandits Problem

**Avrim Blum**                                                                    AVRIM@TTIC.EDU
*Toyota Technological Institute at Chicago, 6045 S. Kenwood Ave., Chicago, IL, 60637*

**Kavya Ravichandran**                                                           KAVYA@TTIC.EDU
*Toyota Technological Institute at Chicago, 6045 S. Kenwood Ave., Chicago, IL, 60637*

**Editors:** Gautam Kamath and Po-Ling Loh

## Abstract

We give nearly-tight upper and lower bounds for the *improving multi-armed bandits* problem. An instance of this problem has $k$ arms, each of whose reward function is a concave and increasing function of the *number of times that arm has been pulled so far*. We show that for any randomized online algorithm, there exists an instance on which it must suffer at least an $\Omega(\sqrt{k})$ approximation factor relative to the optimal reward. We then provide a randomized online algorithm that guarantees an $O(\sqrt{k})$ approximation factor, if it is told the maximum reward achievable by the optimal arm in advance. We then show how to remove this assumption at the cost of an extra $O(\log k)$ approximation factor, achieving an overall $O(\sqrt{k} \log k)$ approximation relative to optimal.

## 1. Introduction

In this paper, we study the problem of improving multi-armed bandits. In this problem, a problem instance consists of $k$ bandit arms (i.e., "pulling" the arm reveals the reward) each with reward that increases the more the arm is pulled. In other words, the payoff is not a function of the *time* at which an arm is pulled but rather of the *number of times it has been pulled so far*, with different arms having (potentially) different increasing functions. Our goal is to maximize the reward we achieve. Some real-world problems captured by this framework include: training multiple learning algorithms, when the performance of an algorithm improves with resources expended and some algorithms are ultimately better than others for the setting at hand (Haussler et al., 1996; Li et al., 2017, 2020); developing new technologies, where investing resources into developing a technology may increase its efficiency and different technologies have different asymptoting utility values (Rivest, 2023); or even the problem of deciding what research area to work in. In each of these examples, each algorithm or technology is represented by one bandit arm, and the reward achieved from pulling the arm increases as it is pulled more. The goal, then, is to find a sequence of arms to pull that will maximize the reward achieved.

If the rewards are *arbitrarily* increasing, then we cannot guarantee much: we could have one arm that gives 0 reward for the first $T/2$ pulls and reward 1 after that, and $k - 1$ arms that are 0 regardless of how many times they've been pulled; the good arm and the bad arms in this case are indistinguishable until it is too late. Thus, prior papers on this problem (Heidari et al., 2016; Patil et al., 2023) consider reward functions that have diminishing returns, i.e., where the difference between two consecutive rewards is non-increasing. The continuous equivalent of this property is concavity. Even with the assumption of diminishing returns and with just two arms, we can see that it is impossible to achieve sublinear additive regret. Suppose one arm linearly increases from 0 to $1/2$

until time $T/2$, and then flattens out, and the other arm increases with the same slope throughout until time $T$. The algorithm has no way of distinguishing which arm it is playing until time $T/2+1$. At this point, if it finds it is playing the first arm, switching to the other arm is worse than continuing to play the first arm. Since there is an $\Omega(T)$ gap between the rewards of the two arms, no algorithm can achieve $o(T)$ regret. Accordingly, we focus on the approximation factor we can achieve, i.e., the ratio of the optimal reward to the reward achieved by the algorithm.

In Patil et al. (2023), the authors study deterministic algorithms and show tight upper and lower bounds of $\Theta(k)$ for the problem. We show that with randomization, we can, in fact, achieve an approximation ratio of $O(\sqrt{k})$ if the algorithm knows the maximum value achieved by the best arm and $O(\sqrt{k}\log k)$ if it does not. We also provide an $\Omega(\sqrt{k})$ lower bound, nearly completely characterizing the achievable approximation guarantees for this problem. Our upper bound is achieved through careful analysis of the recursive structure induced by a natural randomized method for searching through bandit arms. Finally, in Appendix B, we extend our results to the case where the objective is to maximize the maximum reward achieved rather than the total reward achieved, and in Appendix C, we show that even if reward functions are only approximately concave (i.e., concave + noise), we can get nearly the same guarantees.

**Related Work** Our work directly follows up on Heidari et al. (2016); Patil et al. (2023). Heidari et al. (2016) define the setting and problem and shows how to achieve asymptotically sublinear regret[1]. Metelli et al. (2022) study the same asymptotic regret setting but with stochastic rewards. Patil et al. (2023) extends the model in Heidari et al. (2016) to the notion of approximation we consider in our paper. They show that deterministic algorithms must incur an approximation factor of $k$, where $k$ is the number of arms; they also match this with an algorithm that can achieve an $O(k)$ approximation factor.

Another area in which a similar problem has been studied is in the area of online algorithms. Here, the problem is framed as one of a class of *searching* problems (Alpern and Gal, 2003; Gal, 2011). McGregor et al. (2009) consider the problem of a reward (oil, say) present at some depth at one of $k$ locations. Their goal is to minimize the amount of time spent searching; in contrast, in the setup we study, the search time is fixed, and the rewards are gradual, making the objective to maximize reward.

More generally, our work follows in a rich line of work regarding multi-armed bandits (Thompson, 1933; Lattimore and Szepesvári, 2020; Slivkins, 2022). Multi-armed bandits have been well-studied in many different settings. The main idea is that there are many actions an agent could take, but the utility of taking any action is only known (possibly only partially) after taking it. In particular, the nature of the reward function associated with each arm could be stochastic, adversarial, or dependent on some world context. In this work, we study one such setting with rewards that are adversarial up to conforming to the improving, concave structure.

## 2. Preliminaries

We follow the problem specification in Patil et al. (2023). In particular, each instance consists of $k$ arms, where each arm $i$ has an associated monotone increasing reward function $f_i$. The reward from

---

1. More specifically, Heidari et al. (2016) use the fact that for any $k$ bounded and non-decreasing functions $f_i(t)$ and any $\epsilon > 0$, there must exist some $T_\epsilon$ after which each function is within $\epsilon$ of its asymptotic value to achieve asymptotically sublinear regret. In contrast, we will be interested in the case that the adversary can choose the functions $f_i(t)$ after $T$ is fixed.

pulling arm $i$ for the $t^{\text{th}}$ time is $f_i(t)$. These functions are not known to the algorithm in advance; the algorithm only gets to know the current reward by interacting with the arm. Further, the argument $t$ of the function is not the time at which the arm is picked, but rather the number of times the arm has been played. Notice that the best strategy in hindsight is to just identify and pull the arm $i^\star = \arg\max_i \sum_{t=1}^{T} f_i(t)$ the entire time. (Proposition 1 in Heidari et al. (2016)). We will often refer to how things compare to the "optimal" arm; when we say that, we are referring to that arm. For simplicity of notation, we elide the $i$ and refer to this arm as $f^\star(\cdot)$. The reward of playing the optimal arm for the whole time horizon $T$ is $OPT_T$, which we may refer to as $OPT$, and we say playing the best arm for $T'$ steps accrues reward $OPT_{T'}$.

Our goal is to maximize the expected reward $ALG_T$ achieved by our algorithm, i.e., find a sequence of pulls which gives us the maximum cumulative reward over the $T$ time steps (in expectation over internal randomness in the algorithm)[2].

We aim to minimize the approximation factor, which we define as follows.

**Definition 1** *Suppose the expected reward achieved by the algorithm is $ALG$ and the optimal reward achievable is $OPT$. Then, we say that our algorithm achieves a $g$-approximation to the optimal reward if $ALG \geq OPT/g$.*

Not only is approximation factor a natural objective to study (it is fundamental, as defined in the textbook of Borodin and El-Yaniv (2005)), but also it is the metric on which we could hope to understand the problem better. As discussed earlier, achieving sublinear regret in general is impossible, and indeed achieving any non-trivial guarantee with high probability is impossible. To see this, consider a distribution over instances in which there are $k$ arms: $k-1$ of them increase until time $2T/k$ and then flatten, and one arm (chosen randomly) increases linearly throughout. For any algorithm, with probability at least $1/2$, the algorithm will never receive reward better than $2T/k$, because the algorithm can choose at most $k/2$ arms to play more than $2T/k$ times. Thus, achieving a non-trivial guarantee with high probability is not possible.

Following Heidari et al. (2016); Patil et al. (2023), we assume that the reward function for each arm follows the diminishing returns property. Formally,

**Definition 2** *A function $f$ is said to have* diminishing returns *if the following holds for all $t \geq 1$:*

$$f(t+1) - f(t) \leq f(t) - f(t-1).$$

Finally, we assume that $f(0) = 0$ for all of the arms.

## 3. Lower Bound

First, we argue that no randomized algorithm can achieve an $o(\sqrt{k})$ approximation to the optimal reward. In order to show this, we apply Yao's principle (Yao, 1977), i.e., we focus on a distribution over instances that is hard for deterministic algorithms. Theorem 3 presents the distribution over instances and argues the $\sqrt{k}$ lower bound for deterministic algorithms over this distribution. Corollary 4 uses Yao's principle to translate this result to randomized algorithms.

---

2. As is standard in the bandits literature, we define the objective as maximizing cumulative reward. The reader may notice that in some of the motivating applications, the natural goal instead might be to maximize the largest single pull. In Appendix B, we show how all the results from the paper translate to this slightly different objective function.

**Theorem 3** *There exists a distribution over instances of the increasing bandits problem where no deterministic algorithm can achieve expected reward greater than $3\,OPT/\sqrt{k}$.*

**Proof** We begin by defining the following functions. Let $f^\star(t) = t/T \ \forall \ t \in [1, T]$. Let $f(t)$ be the following reward function:

$$f(t) = \begin{cases} \frac{t}{T} & 1 \le t \le \frac{T}{\sqrt{k}} \\ \frac{1}{\sqrt{k}} & \frac{T}{\sqrt{k}} < t \le T \end{cases}.$$

Let us describe a game played by the universe and the algorithm with the following stages.

1. **Universe** sets each of $k$ arms to have reward $f(t)$.

2. **Algorithm** runs for $T$ time steps.

3. **Universe** chooses one arm $i^\star$ uniformly at random and changes its reward function to $f^\star(t)$.

4. **Algorithm** gains $OPT$ reward if it had played the arm replaced with $f^\star$ for at least $T/\sqrt{k}$ steps. Otherwise it keeps the reward it gained originally.

The distribution over instances is exactly specified by the (uniform) distribution over choices for the arm that gets $f^\star(t)$ as its reward function in step (3) above.

First, we explain how the game detailed above is strictly more generous to the algorithm than the original game in which the algorithm has to maximize reward when playing the instance consisting of $k$ arms. Because the sequence of arms **Algorithm** pulls is deterministic, if it has not played arm $i^\star$ more than $T/\sqrt{k}$ times, it cannot change its sequence of pulls based on where $f^\star$ is located. Thus, provided we give the algorithm any extra reward for playing $i^\star$ long enough to identify it, which is done in step 4, we can fix its run before revealing $i^\star$ in step 3.

Now, we argue that the above game ensures that the expected reward of the algorithm, over the randomness in the instance, is at most $3OPT/\sqrt{k}$. We do so by upper bounding the amount an algorithm can achieve while playing this game and lower bounding the value the optimal arm achieves by $T/2$.

Observe that in the $T$ time steps that **Algorithm** runs, at most $\sqrt{k}$ of the arms could have been played for more than $T/\sqrt{k}$ steps. Let us consider the two complementary events:

$$E = \mathbb{I}\left(f^\star \text{ assigned to an arm } \textbf{Algorithm} \text{ played for } > T/\sqrt{k} \text{ steps}\right)$$

$$E^c = \mathbb{I}\left(f^\star \text{ assigned to an arm } \textbf{Algorithm} \text{ played for } \le T/\sqrt{k} \text{ steps}\right)$$

Notice that the maximum reward the algorithm can achieve *conditioned* on event $E^c$ is by playing a single non-optimal arm for all $T$ time steps. We thus upper bound **Algorithm**'s expected reward as follows:

$$ALG = \mathbb{P}\left[E\right] \cdot \text{Reward under event } E + \mathbb{P}\left[E^c\right] \cdot \text{Reward under event } E^c$$

$$\le \frac{\sqrt{k}}{k} OPT + 1 \cdot \left(\frac{1}{2}\frac{T}{T\sqrt{k}}\left(\frac{T}{\sqrt{k}}+1\right) + \left(T - \frac{T}{\sqrt{k}}\right)\frac{1}{\sqrt{k}}\right)$$

$$= \frac{OPT}{\sqrt{k}} + \frac{T}{k} + \frac{T}{\sqrt{k}} - \frac{T}{k} \le \frac{3\,OPT}{\sqrt{k}},$$

since we know that $OPT \ge T/2$. ∎

**Corollary 4** *For any randomized algorithm, there exists an instance for which its approximation factor is at least $\sqrt{k}/3$.*

## 4. Upper Bound

In this section, we present an algorithm with approximation factor $O(\sqrt{k})$, matching the $\Omega(\sqrt{k})$ lower bound in Section 3, when the value of $f^*(T)$ and the value of $T$ itself are given to the algorithm in advance. In Section 5, we show how to remove these assumptions with an $O(\log k)$ factor loss in approximation. To describe the algorithm, we use $m$ to denote the value given to the algorithm that represents $f^*(T)$; we begin by assuming $m = \Theta(f^*(T))$ and then later show how to relax, and then remove, this assumption. Also, we show that in this setting, we can achieve the same approximation factor in terms of $k$ even when the functions do not exactly follow diminishing returns but are close to functions that do (Appendix C).

**Algorithm**   The algorithm chooses an arm $i$ uniformly at random, pulls it so long as its current reward $f_i(t_i)$ is at least $mt_i/T$, where $t_i$ is the number of pulls of arm $i$ so far, and then switches to a different uniformly random arm. (Notice that if $m \leq f^*(T)$ then we will never switch away from the optimal arm.) See Algorithm 1. In Theorem 8, we show that if $m$ is within a constant factor of $f^*(T)$, then the algorithm achieves approximation factor $O(\sqrt{k})$, matching the lower bound from Section 3. We then consider algorithms that aim to adaptively learn a good value of $m$ in Section 5 and give approximation guarantees for the case that $f^*(T)$ and $T$ are not known in advance.

---

**Algorithm 1** Random round robin

---

**1** Parameters $m, T$
**2** $t \leftarrow 0$
**3** $R \leftarrow 0$
   **while** *time not yet expired* **do**
**4**     $i \leftarrow$ Randomly choose arm that has not been chosen so far
**5**     $t_i \leftarrow 0$
        **repeat**
**6**         $t_i \leftarrow t_i + 1$
**7**         $t \leftarrow t + 1$
**8**         pull arm $i$
**9**         $R \leftarrow R + f_i(t_i)$
        **until** $f_i(t_i) < m\,t_i/T$;
   **end**
   **return** $R$

---

To analyze the performance of this algorithm, we begin with the following helpful fact.

**Claim 5**  *If for some constant $c_2 > 1$, $m \in [\frac{1}{c_2} f^*(T), f^*(T)]$ and Algorithm 1 plays arm $i$ for $t_i$ steps, then, from that arm, the algorithm receives total reward at least $\left(\frac{t_i-1}{T}\right)^2 \frac{OPT_T}{2c_2}$.*

**Proof** We know that $i$ is played until $f_i(t_i) < m\, t_i/T$. This gives us that the algorithm's reward is at least:

$$\sum_{\tau=1}^{t_i-1} f_i(\tau) \geq \frac{m}{T} \sum_{\tau=1}^{t_i-1} \tau \geq \frac{(t_i-1)^2}{2} \frac{m}{T}. \tag{1}$$

Since $OPT_T \leq f^*(T)T \leq c_2 mT$, this is at least $\left(\frac{t_i-1}{T}\right)^2 \frac{OPT_T}{2c_2}$ as desired. ∎

Next, define $V(T', k') = \mathbb{E}[$ reward from Algorithm 1 when run for $T' + k'$ steps$]$. Note that this is in the worst case over subsets $\mathcal{A}$ of $k'$ of the original $k$ arms, subject to $\mathcal{A}$ containing the optimal arm. The additional $k'$ steps we give the algorithm in this calculation are simply an analysis tool to help account for the last step on an arm that does not gain sufficient reward but is necessary for the algorithm to know it needs to switch. We will later discuss how to use this recurrence to calculate the quantity we are actually interested in, the expected reward after $T$ steps. We show that $V(T', k')$ satisfies a natural recurrence and then solve that recurrence.

**Lemma 6** *If $m \in [\frac{1}{c_2} f^*(T), f^*(T)]$ then the quantity $V(T', k') = \mathbb{E}[$ reward from Algorithm 1 when there are a total of $k'$ arms and the algorithm is run for time $T' + k']$ satisfies the recurrence below:*

$$V(T', k') \geq \frac{1}{k'} OPT_{T'+k'} + \left(1 - \frac{1}{k'}\right) \min_{0 \leq t \leq T'+k'-1} \left\{ \left(\frac{t}{T}\right)^2 \frac{OPT_T}{2c_2} + V(T' - t, k' - 1) \right\}$$

*where we define $V(T', k') = 0$ for $T' \leq 0, \forall\, k'$.*

**Proof** With probability $\frac{1}{k'}$, Algorithm 1 finds the optimal arm in its first random choice, in which case it receives reward $OPT_{T'+k'}$. If the arm chosen was not the best arm, let the time duration for which reward is received from it be $t + 1$. By Claim 5, the algorithm receives reward at least $\left(\frac{t}{T}\right)^2 \frac{OPT_T}{2c_2}$ while playing that arm, after which the algorithm recurses in a game with one fewer arm (so $k' \leftarrow k' - 1$) and with $T' + k' - t - 1$ time steps to go (so $T' \leftarrow T' - t$). Here, we use the fact that we never discard the optimal arm, which follows from $m \leq f^*(T)$.

Since the value of $t$ depends on the adversarially-chosen function $f_i$, we take a worst case view and lower bound it by the worst possible value of $t \in [0, T' + k' - 1]$, giving us the recurrence shown. ∎

Now, we analyze this recurrence to get a closed form bound on its value in terms of the amount of reward received by the optimal strategy played for $T$ steps.

**Lemma 7** *Suppose we run Algorithm 1 with a parameter $m \in [\frac{1}{c_2} f^*(T), f^*(T)]$. Then the recurrence given in Lemma 6 evaluates as follows: for all $T'$, $V(T', k') \geq \left(\frac{T'}{T}\right)^2 \frac{OPT_T}{2c_2} \cdot \frac{1}{\sqrt{k'}}$. In particular, $V(T, k) \geq \frac{OPT_T}{2c_2 \cdot \sqrt{k}}$.*

**Proof** We prove the desired statement by induction on $T'$ and $k'$.

**Base Case:** We start by considering the base cases $V(1, k')$ and $V(T', 1)$. For the first case, notice that if the very first pull of an arm produces reward less than $m/T$, then the algorithm will immediately choose a new arm for its next pull. Therefore, the algorithm is guaranteed to at least once pull an arm with reward at least $m/T \geq \frac{OPT_T}{c_2 T^2} \geq \left(\frac{1}{T}\right)^2 \frac{OPT_T}{2c_2}$. For the second case, the instance only has one arm, which must be the optimal arm, so it clearly receives $OPT_{T'+1} \geq OPT_{T'} \geq \left(\frac{T'}{T}\right)^2 \frac{OPT_T}{2c_2}$ reward, where the last inequality follows from diminishing returns and $2c_2 > 1$.

**Inductive Assumption:** Assume that $\forall\, T'' < T'$ and $k'' < k', V(T'', k'') \geq \left(\frac{T''}{T}\right)^2 \frac{OPT_T}{2c_2} \cdot \frac{1}{\sqrt{k''}}$.

**Induction:** First, observe that the right-hand side of the recurrence in Lemma 6 can be lower bounded by replacing $OPT_{T'+k'}$ with $OPT_{T'}$:

$$V(T', k') \geq \frac{1}{k'}OPT_{T'} + \left(1 - \frac{1}{k'}\right) \min_t \left\{ \left(\frac{t}{T}\right)^2 \frac{OPT_T}{2c_2} + V(T' - t, k' - 1) \right\}. \qquad (2)$$

From the inductive assumption, we have:

$$V(T', k') \geq \frac{1}{k'}OPT_{T'} + \left(1 - \frac{1}{k'}\right) \min_t \left\{ \left(\frac{t}{T}\right)^2 \frac{OPT_T}{2c_2} + \frac{\left(\frac{T'-t}{T}\right)^2 \frac{OPT_T}{2c_2}}{\sqrt{k'-1}} \right\} \qquad (3)$$

$$\geq \frac{1}{k'}OPT_{T'} + \left(1 - \frac{1}{k'}\right) \min_t \left\{ \left(\frac{t}{T}\right)^2 \frac{OPT_T}{2c_2} + \frac{1}{\sqrt{k'}} \left(\frac{T'-t}{T}\right)^2 \frac{OPT_T}{2c_2} \right\} \qquad (4)$$

Next, let us compute the minimum desired. We take the derivative with respect to $t$ and set it to 0 (note that at $t = 0$, the derivative is negative, and at $t = T$, the derivative is positive, so the minimum must lie in the middle):

$$\frac{2t}{T^2} \frac{OPT_T}{2c_2} + \frac{OPT_T}{2c_2} \frac{1}{\sqrt{k'}} \cdot 2 \left(\frac{T'-t}{T}\right) \frac{-1}{T} = 0 \qquad (5)$$

$$t = \frac{1}{\sqrt{k'}}(T' - t) \Leftrightarrow t = \frac{T'}{\sqrt{k'} + 1} \qquad (6)$$

Here, note that $\frac{T'}{\sqrt{k'}+1} < T'$. Plugging this back in, we get:

$$\min_t \left\{ \left(\frac{t}{T}\right)^2 \frac{OPT_T}{2c_2} + \frac{1}{\sqrt{k'}} \left(\frac{T'-t}{T}\right)^2 \frac{OPT_T}{2c_2} \right\} = \frac{OPT_T}{2c_2} \left(\frac{T'}{T}\right)^2 \frac{1}{\sqrt{k'}+1} \qquad (7)$$

Finally, we can plug this back to get the guarantee:

$$\frac{1}{k'}OPT_{T'} + \left(1 - \frac{1}{k'}\right) \frac{OPT_T}{2c_2} \left(\frac{T'}{T}\right)^2 \frac{1}{\sqrt{k'}+1}$$

$$\geq \frac{OPT_T}{2c_2} \left(\frac{T'}{T}\right)^2 \left(\frac{1}{k'} + \left(1 - \frac{1}{k'}\right) \frac{1}{\sqrt{k'}+1}\right)$$

$$= \frac{OPT_T}{2c_2} \left(\frac{T'}{T}\right)^2 \left(\frac{\sqrt{k'}+k'}{k'(\sqrt{k'}+1)}\right) = \frac{OPT_T}{2c_2} \left(\frac{T'}{T}\right)^2 \frac{1}{\sqrt{k'}}.$$

∎

With this fact in hand, let us return our attention to computing the reward for the algorithm run for $T$ steps. For this, we must consider first the value of $V(T-k, k)$, which we get in terms of $OPT_{T-k}$. Thus, we next compare that to $OPT_T$, which is the actual optimal value the algorithm must compete with.

**Theorem 8** *If $m \in [\frac{1}{c_2} f^*(T-k), f^*(T-k)]$ and $T \geq 2k$, then Algorithm 1, run with $T - k$ as its parameter "$T$", receives expected reward at least $\frac{OPT_T}{8c_2\sqrt{k}}$ in $T$ steps.*

**Proof** Lemma 7 shows that Algorithm 1, run for $T + k$ steps, receives expected reward at least $\frac{OPT_T}{2c_2 \cdot \sqrt{k}}$. Therefore, if we run the algorithm with $T - k$ as its value of "$T$", then in $T$ steps, where $T \geq 2k$, it will receive expected reward at least

$$\frac{OPT_{T-k}}{2c_2 \cdot \sqrt{k}} \geq \left(\frac{T-k}{T}\right)^2 \frac{OPT_T}{2c_2 \cdot \sqrt{k}} \geq \frac{OPT_T}{8c_2 \cdot \sqrt{k}}.$$

∎

## 5. Removing Assumptions

In the previous section, we assumed knowledge of $f^\star(T)$, the maximum value attained by the best arm, to within a constant factor. In this section, we give an extension of our previous algorithm that essentially makes an educated guess of this value based on some initial exploration. In order to do so, we now require that $T > 4k$. This is so that we can conduct an initial exploration phase for the first half of the time steps and then conduct our previous algorithm for the second half based on our "learned" guess for $m$, which we shall call $\hat{m}$.

**Main Ideas For Learning $\hat{m}$** We spend half of the time available to us "learning" the parameter $m$ with which we will run Algorithm 1, as detailed in Algorithm 2. In the other half of the time available, we accrue reward through running Algorithm 1. In order to learn the parameter, we follow the following main steps: first, we pull each arm $T/(2k)$ times. We use the last two values to determine an upper and lower bound for the maximum possible value achievable by *that* arm, $f_i(T)$, (lines 1-3 of Algorithm 2, analyzed in Lemma 9). We then narrow down a range inside which the maximum of the *best* arm ($f^\star(T)$) must lie (lines 4-5 of Algorithm 2, analyzed in Lemma 10). Finally, we randomly choose a value in that interval that, with probability $\Omega(\frac{1}{\log k})$, is within a factor of 2 of $f^\star(T)$ (lines 6-7 of Algorithm 2, analyzed in Theorem 11). Once we have that, we can use the guarantee from Theorem 8 to analyze the output of Algorithm 1 run with the learned parameter $m$.

Technically, we will learn an estimate of $f^*(\frac{T}{2} - k)$ (rather than $f^*(T)$) since that will be the value of "$T$" given to Algorithm 1; for this reason, we provide a parameter $T_{pred}$ to the algorithm below. Note that $f^*(T)$ and $f^*(\frac{T}{2} - k)$ differ by only a constant factor.

---

**Algorithm 2** Find $\hat{m}$

---

**Input:** parameter $T_{\text{pred}}$

**for** *each arm $i$* **do**

1    pull arm $T/(2k)$ times

2    $\hat{m}_L^{(i)} \leftarrow f_i\left(\frac{T}{2k}\right)$

3    $\hat{m}_U^{(i)} \leftarrow f_i\left(\frac{T}{2k}\right) + \left(f_i\left(\frac{T}{2k}\right) - f_i\left(\frac{T}{2k} - 1\right)\right)\left(T - \frac{T}{2k}\right)$

**end**

4 $L \leftarrow \frac{1}{2} \max_i \hat{m}_L^{(i)}$

5 $U \leftarrow \max_i \hat{m}_U^{(i)}$

6 Define uniform probability distribution $p$ over $\{-\log(T/T_{\text{pred}}), \ldots, 0, 1, 2, \ldots, \log(U/L)\}$

7 Draw $j \sim p$

**return** $L \cdot 2^j$

---

## 5.1. Computing Range for a Fixed Arm

Here, we describe how we compute the range in which the maximum lies for a fixed arm, i.e., we compute upper and lower bounds on $f_i(T)$ based on an exploration phase on $f_i$.

**Lemma 9** *The procedure detailed in lines 1-3 of Algorithm 2 provides, for a fixed arm with reward function $f$, a range $\mathcal{R} := [\hat{m}_L, \hat{m}_U]$ such that $f(T) \in \mathcal{R}$ and $\hat{m}_U \leq 2k\,\hat{m}_L$.*

**Proof** We use the diminishing returns property to show both parts. First, by the fact that the reward functions are increasing, it is clear that $f(T) \geq f(T/(2k))$. Then, due to the diminishing returns property:

$$f(T) = f\left(\frac{T}{2k}\right) + \sum_{n=1}^{T-\frac{T}{2k}} \left\{ f\left(\frac{T}{2k} + n\right) - f\left(\frac{T}{2k} + n - 1\right) \right\} \tag{8}$$

$$\leq f\left(\frac{T}{2k}\right) + \left(T - \frac{T}{2k}\right)\left(f\left(\frac{T}{2k} + 1\right) - f\left(\frac{T}{2k}\right)\right) \tag{9}$$

$$\leq f\left(\frac{T}{2k}\right) + \left(T - \frac{T}{2k}\right)\left(f\left(\frac{T}{2k}\right) - f\left(\frac{T}{2k} - 1\right)\right). \tag{10}$$

Next, we show that $\hat{m}_U \leq 2k\hat{m}_L$, which follows immediately from the diminishing returns property. In particular, $\hat{m}_U = f(\frac{T}{2k}) + (f(\frac{T}{2k}) - f(\frac{T}{2k} - 1))(T - \frac{T}{2k}) \leq f(\frac{T}{2k}) + \frac{2k}{T}f(\frac{T}{2k})(T - \frac{T}{2k}) = f(\frac{T}{2k}) + (2k-1)f(\frac{T}{2k}) = 2k\,\hat{m}_L$. ∎

## 5.2. Upper and Lower Bounds on $f^\star(T)$ From Exploration

Now, for each arm, we have a range within which its maximum must lie. Next, we combine these bounds to get a relatively small interval in which we can be sure $f^\star(T)$ lies.

**Lemma 10** *Define $L := \frac{1}{2} \max_i \hat{m}_L^{(i)}$ and $U := \max_i \hat{m}_U^{(i)}$. Then, $f^\star(T) \in [L, U]$, and $\frac{U}{L} \leq 4k$.*

**Proof** First, observe that by diminishing returns, we have that:

$$f^\star(T)\,T \geq \sum_{t=1}^{T} f^\star(t) = \max_i \sum_{t=1}^{T} f_i(t) \geq \sum_{t=1}^{T} f_p(t) \quad \text{where } p := \arg\max_i f_i\left(\frac{T}{2k}\right) \tag{11}$$

$$\geq f_p\left(\frac{T}{2k}\right)\frac{T}{2} = \hat{m}_L^{(p)}\frac{T}{2} \Leftrightarrow f^\star(T) \geq \frac{\hat{m}_L^{(p)}}{2} = L\,. \tag{12}$$

Next, since $f^\star(T) \leq f^\star(\frac{T}{2k}) + (T - \frac{T}{2k})(f^\star(\frac{T}{2k}) - f^\star(\frac{T}{2k} - 1))$ (due to diminishing returns), we also have that $f^\star(T) \leq \max_i f^{(i)}(\frac{T}{2k}) + (T - \frac{T}{2k})(f^{(i)}(\frac{T}{2k}) - f^{(i)}(\frac{T}{2k} - 1)) = \max_i \hat{m}_U^{(i)} = U$.

Now that we have that $m \in [L, U]$, we verify the size of the interval. In particular, for some index $i'$ we have $U = \hat{m}_U^{(i')} \leq 2k\hat{m}_L^{(i')} \leq 4kL$, where the first inequality follows from Lemma 9. ∎

### 5.3. Reward Approximation Guarantee With Learned Parameter

Putting the pieces together, we show that choosing $\hat{m}$ randomly according to the defined uniform distribution in line 6 of Algorithm 2 allows us to get reward as before with the loss of a factor only logarithmic in $k$.

**Theorem 11** *When $T > 4k$, picking the parameter $m$ required by Algorithm 1 using the procedure detailed in Algorithm 2 with parameter $T_{pred} = \frac{T}{2} - k$ and running Algorithm 1 with that value $\hat{m}$ achieves expected reward $ALG = \Omega(OPT/(\sqrt{k}\log k))$.*

**Proof** To show this, we analyze the three parts of the algorithm. First, we analyze lines 1-3 of Algorithm 2 in Lemma 9. Then, we analyze the range $\mathcal{R} = [L, U]$ in Lemma 10. With these in hand, we show that we can find a constant factor approximation to $f^\star(T)$ with good enough probability that the overall expected reward only gets worse by a logarithmic factor.

First, let us untangle the relationship between $f^\star\left(\frac{T}{2} - k\right)$, which we must use in calling Algorithm 1 to get the guarantee as per Lemma 7, and $f^\star(T)$, which we know we can approximate from the previous two lemmas. Observe that due to increasing nature of $f^\star$, $f^\star(T) \geq f^\star\left(\frac{T}{2} - k\right) \geq f^\star\left(\frac{T}{4}\right)$. Further, due to the diminishing returns property, we have $f^\star\left(\frac{T}{4}\right) \geq f^\star(T)\frac{T}{4}/T = \frac{f^\star(T)}{4}$. Thus, since $U \geq f^\star(T) \geq L$, $U \geq f^\star(T/4) \geq L/4$. Note that $\log(T/T_{\text{pred}}) \leq \log(4)$.

We focus on the selection of $\hat{m}$. We divide up the interval $\mathcal{R}$ by doubling $L$, i.e., the $i^{th}$ interval endpoint is $L \cdot 2^{i-1}$, for $i \in \{-2 \leq -\log(T/T_{\text{pred}}), \ldots, 1, 2, \ldots, \log\left(\frac{U}{L}\right) \leq \log(4k)\}$. We know that $f^\star(T/4)$ must lie in one of these intervals, so if we pick the endpoint, we will be within a factor of 2 as desired above. In particular, let us choose $L \cdot 2^i$ uniformly over the set. The probability of choosing $L \cdot 2^i$ is exactly $\frac{1}{3+\log(U/L)}$ which is at least $\frac{1}{3+\log(4k)}$. Thus, with probability $\geq \frac{1}{3+\log(4k)}$, we choose $\hat{m}$ such that $\hat{m} \leq f^\star(T/2 - k) \leq 2\hat{m}$. When $\hat{m} \leq f^\star(T/2 - k) \leq 2\hat{m}$, we have that $\hat{m} \leq f^\star(T) \leq 8\hat{m}$, since $f^\star(T)/4 \leq f^\star(T/2 - k) \leq f^\star(T)$ Thus, we can apply Lemma 7 with $c_2 = 8$, and $T' = T/2 - k$ to get that:

$$ALG_{T/2} = V\left(\frac{T}{2} - k, k\right) \geq \frac{OPT_{T/2-k}}{2c_2\sqrt{k}} \geq \left(\frac{\frac{T}{2} - k}{T}\right)^2 \frac{OPT_T}{2c_2\sqrt{k}} \geq \frac{OPT_T}{256\sqrt{k}}\,.$$

Putting these together, the expected reward is at least $\frac{1}{3+\log(4k)}$ times the reward as described above, giving us that the reward here is at least:

$$\frac{1}{3+\log(4k)} \cdot \frac{OPT_T}{256\sqrt{k}} \geq \Omega\left(\frac{OPT}{\sqrt{k}\log(k)}\right) .$$

∎

### 5.4. Removing Dependence on Having To Know $T$

Finally, we describe how to use a standard doubling trick to remove dependence on knowing $T$, the time horizon, a priori. At a high level, we start with a guess, $T' = T_0 = 4k$, pretend that is all the time we have, and run the combined exploration-exploitation algorithm with it. Then, if it turns out we have not run out of time, we set $T' \leftarrow 2 \cdot T'$ and repeat the same process. At a high level, our argument will show that if we have a "good" $T'$, i.e., one that is within a constant factor of the true $T$, then the reward received when running $T'/2$ steps of Algorithm 2 followed by $T'/2$ steps of Algorithm 1 will be within a constant factor of the reward received if we were to play until time $T$. Overall, in the exploration phase, we use the algorithm given in Algorithm 2, but a key change is that in line 5, we instead set $U = 4 \max_i \hat{m}_U^{(i)}$. We present the algorithm and result here and defer the proof to Appendix A.

---

**Algorithm 3** Unknown Time Wrapper

---
**1** $R \leftarrow 0$
**2** Guess $T' \leftarrow T_0$
  **while** *True* **do**
**3**  Try:
      $T_{\text{pred}} = T'/2 - k$
      $\hat{m} \leftarrow$ modified Algorithm 2 for $T'/2$ steps with $T_{\text{pred}}$ as above
      `// in line 5 of Algorithm 2,` $U = 4 \cdot \max_i \hat{m}_U^{(i)}$
      $R \leftarrow R +$ reward from Algorithm 1 for $T'/2 - k$ steps
**4**  Except:
      **return** $R$
**5**  $T' \leftarrow 2 \cdot T'$
  **end**

---

**Lemma 12** *Suppose we run the procedure described in Algorithm 3. If $T > 4k$, then the reward achieved is at least* $\frac{OPT_T}{8192\sqrt{k}\log(128k)}$.

### Acknowledgments

This work was supported in part by the National Science Foundation under grants CCF-2212968 and ECCS-2216899 and by the Simons Foundation under the Simons Collaboration on the Theory of Algorithmic Fairness.

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

## Appendix A. Proof of Lemma 12

**Proof** Let $T'' = (2^i - 1)T_0$ be such that $T'' \leq T \leq 2T''$, which implies that the last $T'$ with which we we run Algorithms 2 and 1 is $2^{i-1}T_0$. Note that $T' \leq T'' \leq T \leq 2T'' \leq 4T'$ Then, we consider what we gain when we spend $T'/2$ exploring and $T'/2$ exploiting. Our analysis of the reward from Algorithm 1 will assume we start from 0 pulls in the exploitation phase, but in practice we are already higher than that.

We show that the reward achieved running Algorithm 1 for $T'/2$ steps is a constant fraction of $OPT_T$.

Each arm has been pulled $T'/(2k)$ times in the exploration phase. We know that $f^\star(T') \in [\frac{1}{2} \max_i \hat{m}_L^{(i)}, \max_i \hat{m}_U^{(i)}]$. As before, we actually run Algorithm 1 with its parameter set to $T'/2 - k$. Due to the diminishing returns property, we have that $\frac{f^\star(T')}{f^\star(T'/2-k)} \frac{f(T'')}{f(T')} \frac{f^\star(T)}{f^\star(T'')} \leq \frac{T'}{T'/2-k} \frac{T''}{T'} \frac{T}{T''} \leq 4 \cdot 2 \cdot 2 = 16$. Thus, $f^\star(T) \leq 2f^\star(T'') \leq 4f^\star(T') \leq 4 \max_i \hat{m}_U^{(i)}$, and $c_2 = 16$. Now, the extent of the interval is $128k$, so the probability of choosing $\hat{m}$ that is within a constant factor of $f^\star(T'/2 - k)$ is $\frac{1}{\log 128k}$. Now, we simply compute the reward achieved by the algorithm by applying Lemma 7. The algorithm plays for $T'/2 - k$ steps, while OPT plays for $T$ steps. In particular:

$$
\begin{aligned}
ALG_{T'/2} = V\left(\frac{T'}{2} - k, k\right) &\geq \frac{OPT_{T'/2-k}}{2c_2} \frac{1}{\sqrt{k}} \geq \left(\frac{\frac{T'}{2} - k}{T'}\right)^2 \frac{OPT_{T'}}{2c_2} \frac{1}{\sqrt{k}} \\
&\geq \left(\frac{1}{2} - \frac{1}{4}\right)^2 \frac{OPT_{T'}}{2c_2} \frac{1}{\sqrt{k}} \geq \frac{1}{16} \frac{T'^2}{T^2} \frac{OPT_T}{2c_2} \frac{1}{\sqrt{k}} \\
&\geq \frac{1}{16 \cdot 16} \frac{OPT_T}{2c_2} \frac{1}{\sqrt{k}}.
\end{aligned}
$$

Plugging in the computed value for $c_2$ and considering the probability of choosing the correct $\hat{m}$, we get $ALG_{T'/2} \geq OPT/(8192 \cdot \sqrt{k} \log(128\,k))$. ∎

## Appendix B. Maximum Reward Objective

In certain situations, instead of trying to maximize the sum over rewards, we wish to just maximize the maximum reward achieved in a single step. For instance, if we are training different machine learning models, we care about the best performance the model can achieve, not the cumulative performance over training iterations. Our upper bound results from the main paper all hold up to constant factors for this objective, since we exploit the relationship between the maximum value of a function with diminishing returns and the area under it in many places. Our lower bound result also holds but requires a bit of care. In this section, we formally argue the small changes we make in order for the results in the main paper to translate to this different objective.

### B.1. Lower Bound

We use the same construction and game as before but now directly argue about the pull with maximum reward rather than the sum. For this we have that the algorithm's expected reward can be upper

bounded as follows:

$$ALG = \mathbb{P}[E] \cdot \text{Reward under event } E + \mathbb{P}[E^c] \cdot \text{Reward under event } E^c \tag{13}$$

$$\leq \frac{\sqrt{k}}{k}1 + 1 \cdot \frac{1}{\sqrt{k}} = \frac{2}{\sqrt{k}} \tag{14}$$

$$= \frac{2\,OPT}{\sqrt{k}}\,. \tag{15}$$

So we have that the competitive ratio is at least $\Omega(\sqrt{k})$ as before.

### B.2. Upper Bound

The following facts allow us to directly translate the theorem statements to equivalent ones under this new objective function.

**Fact B.1** *For a given function $f$ satisfying the diminishing returns property, we can relate the area under the function to its maximum value $m$ as follows:*

$$mT \geq \sum_{t=1}^{T} f(t) \geq \frac{mT}{2}\,.$$

**Fact B.2** *Out of a given set of functions $\mathcal{F}$, each satisfying the diminishing returns property, define $f_1^\star := \arg\max_{f \in \mathcal{F}} f(T)$ and $f_2^\star := \arg\max_{f \in \mathcal{F}} \sum_{t=1}^{T} f(t)$. Then, we have that $f_1^\star(T) \leq 2f_2^\star(T)$.*

**Proof** We have that:

$$f_2^\star(T) \cdot T \geq \sum_{t=1}^{T} f_2^\star(t) \tag{16}$$

$$\geq \sum_{t=1}^{T} f_1^\star(t) \tag{17}$$

$$\geq \frac{f_1^\star(T) \cdot T}{2}\,. \tag{18}$$

∎

Thus, we use that $OPT_T \geq \frac{f_1^\star(T) \cdot T}{2}$.

**Fact B.3** *The maximum reward achieved in a single pull over the course of running the algorithm is at least the average reward per pull over the pulls of the algorithm, which is exactly the sum of rewards achieved by the algorithm, referred to as $ALG$ in the main body of the paper, divided by $T$.*

Putting these three facts together, we have:

1. **Translating Theorem 8.** The maximum pull by the algorithm is at least $ALG/T$, which by the theorem is at least $OPT/(8c_2 T\sqrt{k}) \geq f_1^\star(T)/(16c_2\sqrt{k})$. Thus, the $O(\sqrt{k})$ competitive ratio still holds.

2. **Translating Theorem 11.** By the same argument above, we divide both sides of the result by $T$ and get that the maximum pull by the algorithm is at least $1/O(\sqrt{k}\log k)$ fraction of the maximum pull achievable.

3. **Translating Lemma 12.** By the same argument as before, we divide both sides by $T$ to get that the maximum pull of the algorithm is at least $1/(16384\sqrt{k}\log(128k))$ fraction of the maximum pull achievable.

## Appendix C. Extension to noisy rewards

### C.1. Extension of Algorithm 1

Now, suppose that when we pull an arm, instead of getting the exact value of $f_i(t_i)$, we get a reward value $\hat{f}_i(t_i) \in [(1-\epsilon)f_i(t_i), (1+\epsilon)f_i(t_i)]$. We now wish to achieve cumulative reward that competes with the reward achieved by the best policy. The best policy is no longer necessarily a single arm, since the hat reward functions could be slightly non-monotone. However, it suffices to compare to the policy that plays the arm corresponding to the best pre-corruption arm $f^\star$. This is for the following reason. Let $\pi$ be the policy that plays only $f^\star$, and let $\hat{\pi}$ be the policy that achieves the optimal reward in the $\hat{f}$ problem. Define $V(\cdot)$ to be the value of the policy in the argument under the original problem and $\hat{V}(\cdot)$ be the value of the policy in the argument in the hat problem. Then:

$$OPT_T = \hat{V}(\pi) \geq (1-\epsilon)V(\pi) \quad \text{reward at each step in hat problem at least } 1-\epsilon \text{ reward in original problem} \tag{19}$$

$$\geq (1-\epsilon)V(\hat{\pi}) \quad \text{optimality of } \pi \text{ for } V \tag{20}$$

$$\geq \frac{1-\epsilon}{1+\epsilon}\hat{V}(\hat{\pi}) \quad \text{reward at each step in original problem} \geq \frac{1}{1+\epsilon} \text{ of reward in hat problem} \tag{21}$$

Thus, in the rest of this section until the end, we compare to $\hat{V}(\pi) = OPT_T$, and then finally we can lower bound that by the quantity in Eqn. 21.

We show that the same algorithm works with a minor modification. In particular, instead of switching arms away from arm $i$ when $f_i(t_i) < m \cdot t_i/T$, we now switch away from arm $i$ when $\hat{f}_i(t_i) < (1-\epsilon)mt_i/T$. Note that the algorithm therefore must have knowledge of the value of $\epsilon$.

We must show two things: first, that as before, we never switch off the best arm; second, that similar to before, playing an arm for time $t_i$ accrues at least a constant fraction of $\left(\frac{t_i-1}{T}\right)^2 \frac{OPT_T}{2c_2}$. The first can be seen as follows: for an arm $f_i$, as long as $f_i(t_i) \geq t_i m/T$, we have $\hat{f}_i(t_i) \geq (1-\epsilon)f_i(t_i) \geq (1-\epsilon)t_i m/T$, so we will continue playing that arm. In particular, this holds for the best arm.

For the second, we prove a variant of Claim 5:

**Claim 13** *Suppose $m \in [\frac{1}{c_2}f^\star(T), f^\star(T)]$. In the noisy case, if arm $i$ is played by Algorithm 1 with the adaptation mentioned above for $t_i$ steps, then the algorithm receives total reward at least $\left(\frac{t_i-1}{T}\right)^2 \frac{OPT_T}{2c_2} \cdot \frac{1-\epsilon}{1+\epsilon}$ in those steps.*

**Proof** We know that arm $i$ is played until $\hat{f}_i(t_i) < (1 - \epsilon)mt_i/T$. This means the total reward from this arm is:

$$\sum_{\tau=1}^{t_i-1} \hat{f}_i(t_i) \geq \frac{(1-\epsilon)\,m}{T} \sum_{\tau=1}^{t_i-1} \tau \tag{22}$$

$$\geq \frac{(1-\epsilon)\,m}{T} \frac{(t_i-1)^2}{2}. \tag{23}$$

Since $OPT_T \leq \hat{f}^\star(T)\,T \leq (1+\epsilon)f^\star(T)\,T \leq (1+\epsilon)c_2\,m\,T$, the right hand side above is at least $\frac{1-\epsilon}{1+\epsilon}\left(\frac{t_i-1}{T}\right)^2 \frac{OPT_T}{2c_2}$. ■

Finally, we must incorporate this into the recursion of Lemma 6 and then change Lemma 7 slightly. In particular:

$$V(T',k') \geq \frac{1}{k'}OPT_{T'+k'} + \left(1 - \frac{1}{k'}\right) \min_{0 \leq t \leq T'+k'-1} \left\{ \frac{1-\epsilon}{1+\epsilon}\left(\frac{t}{T}\right)^2 \frac{OPT_T}{2c_2} + V(T'-t, k'-1) \right\} \tag{24}$$

where we define $V(T',k') = 0$ for $T' \leq 0$, $\forall\,k'$.

Finally, we aim now to show that $V(T',k') \geq \frac{1-\epsilon}{1+\epsilon}\left(\frac{T'}{T}\right)^2 \frac{OPT_T}{2c_2} \frac{1}{\sqrt{k'}}$. The base cases go through as before, since $\frac{1-\epsilon}{1+\epsilon}$ times the lower bound is only smaller than the lower bound for which we have shown it. Further, let the inductive assumption be $V(T'',k'') \geq \frac{1-\epsilon}{1+\epsilon}\left(\frac{T''}{T}\right)^2 \frac{OPT_T}{2c_2} \frac{1}{\sqrt{k''}}$, $\forall\,T'' < T'$, $k'' < k'$. Now, we can lower bound the right hand side of Eqn. 24 by:

$$\frac{1-\epsilon}{1+\epsilon}\left(\frac{1}{k'}OPT_{T'} + \left(1 - \frac{1}{k'}\right) \min_t \left\{ \left(\frac{t}{T}\right)^2 \frac{OPT_T}{2c_2} + \frac{1}{\sqrt{k'}}\left(\frac{T'-t}{T}\right)^2 \frac{OPT_T}{2c_2} \right\}\right).$$

Now, we can analyze everything multiplied by $(1-\epsilon)$ exactly as before, and we would prove the following statement:

**Theorem 14** *Suppose that when we pull an arm $i$ for the $t_i$th time, instead of getting value $f_i(t_i)$, we get $\hat{f}_i(t_i) \in (1\pm\epsilon)f_i(t_i)$. Suppose we run the adapted version of Algorithm 1 as discussed above with a parameter $m \in [\frac{1}{c_2}f^*(T), f^*(T)]$. Then the recurrence given in Eqn. 24 evaluates as follows: for all $T'$, $V(T',k') \geq \frac{1-\epsilon}{1+\epsilon}\left(\frac{T'}{T}\right)^2 \frac{OPT_T}{2c_2} \cdot \frac{1}{\sqrt{k'}}$. In particular, $V(T,k) \geq \frac{1-\epsilon}{1+\epsilon}\frac{OPT_T}{2c_2\cdot\sqrt{k}} \geq \left(\frac{1-\epsilon}{1+\epsilon}\right)^2 \frac{\hat{V}(\hat{\pi})}{2c_2\cdot\sqrt{k}}$.*

Finally, we can incorporate the extra switching steps to get a theorem like before with the same $\left(\frac{1-\epsilon}{1+\epsilon}\right)^2$ multiplicative factor.

## C.2. Extension of Algorithm 2

Now, we define:

$$\hat{m}_L = \frac{\hat{f}\left(\frac{T}{2k}\right)}{1+\epsilon} \qquad \hat{m}_U = 2k \cdot \frac{\hat{f}\left(\frac{T}{2k}\right)}{1-\epsilon}.$$

It is clear by our assumptions on $\hat{f}$ and diminishing returns that $f(T)$ lies between $\hat{m}_L$ and $\hat{m}_U$. Further, it is clear that $\hat{m}_U \leq 2k \frac{1+\epsilon}{1-\epsilon} \hat{m}_L$. The rest of the arguments go through, and now the approximation factor is $O\left(\left(\frac{1-\epsilon}{1+\epsilon}\right)^2 \sqrt{k} \log\left(\frac{1+\epsilon}{1-\epsilon} k\right)\right)$.

Note that this definition of $\hat{m}_U$ could have also been used in the noiseless case with the same guarantee. However, in practice we may get a better interval if we do what we originally had in the algorithm.

