# OpenReview forum: "Nearly-tight Approximation Guarantees for the Improving Multi-Armed Bandits Problem"
_algorithmiclearningtheory.org/ALT/2025/Conference — ALT 2025_

### Official Review · Reviewer_maSC · 2024-10-14

**Rating:** 5
**Confidence:** 4

**Review:**

This paper studies the improving (rising) bandit problem, in which each arm’s reward follows an arbitrary increasing function with diminishing returns.  Several variations of this have been studied previously, and what distinguishes this one is (i) considering instances whose worst-case nature may depend on the time horizon T, and (ii) allowing randomization rather than only deterministic strategies.  The lower bound shows that the regret must be $\Omega(\sqrt{k})$ times the best single action, and the upper bound matches this (when the best total reward is known to within a constant factor) or gets within a log factor (otherwise).

The results seem to be good additions to this line of works, but I also have some not-so-minor concerns mentioned below.  I am unsure about acceptance, as I would ideally (if available) ask for a “major revision with second round review”, but I wouldn’t mind if the paper were accepted.

Main comments:
1) The paper does not do a good job of highlighting the distinction between average regret and high-probability regret.  It is absolutely essential for the results given – it looks like even asking for a probability-$\frac{1}{2}$ regret bound would mean we’re back to $O(k)$ suboptimality rather than $O(\sqrt{k})$, because the hard instance could be such that information about the optimal arm is only learned after time $\frac{2T}{k}$.  Along similar lines, the upper bound in Section 5 is based on identifying an $\Omega\big( \frac{1}{\log k} \big)$ probability good event, which means that the algorithm might actually fail most of the time.  The average regret notion considered may be of some theoretical interest (perhaps less practical interest?) but it does stand out as a weakness, and it really needs to be highlighted better.  This is made worse by the fact that Section 2 never explicitly writes down the performance goal mathematically (which should be a displayed equation with a a clear E[.] operation etc.) and these subtle issues are really hidden.
2) The lower bound proof seems to be very similar to that of Patil et al., but I don’t see any mention of that in the proof.  I find this rather troubling, as it makes me uncertain about which methods in the entire paper are more novel or less novel.
3) The main assumptions are also quite hidden in the text.  I suggest a LaTeX “Assumption” environment clearly stating the diminishing returns, f(0)=0, and anything else you assume.
4) Sometimes the mathematical steps themselves can also be clearer.  For example, $OPT_{T’} \ge \big(\frac{T’}{T}\big)^2 OPT_T$ should be stated as a lemma *with proof*, and cross-referenced whenever used (it’s not immediate to see this just “from diminishing returns”).
5) Other parts of the writing can also be improved.  For example, Lemma 6 and the paragraph before it are quite confusing and events/quantities are written in an overly wordy way, sometimes in an incomplete or inconsistent manner.  It’s better to use mathematical definitions and symbols.

Minor comments:
- The abstract and introduction mention that any algorithm must be $\sqrt{k}$ away from the “optimal reward”.  This is ambiguous, because “optimal reward” could be taken to mean the reward obtained by some actual algorithm for your problem.  I suggest wording to the effect of “total reward of the best single action”.
- I suggest merging Footnote 1 into the main text.
- The final sentence of Section 1 is very generic.  I think a better job can be done of highlighting the other bandit literature.
- p6: After “…from diminishing returns” you can also write “…and $2c_1 > 1$” to avoid confusion with two different steps being applied at once.  You can similarly give a hint about the second-last line in this proof (just before Theorem 8).
- Suggest brackets around the summand in (8)
- At the end of Lemma 9’s proof I don’t know why you write X+(2k-1)X instead of 2kX.
- In Appendix C you should point out (if not already done) that you assume $\epsilon$ to be known to the algorithm.
- In Claim 13 you should say *adapted version* of Algorithm 1
- I think the word “noisy” should be mentioned in the main text, not just Appendix C.  Stating that you extend to noisy settings sounds stronger than just stating that you extend to cases without precisely diminishing returns.

**Paper Award:**

No

---

> ### Author Response · Authors · 2024-11-25
>
> 1.	The average competitive ratio notion considered is standard in the randomized online algorithms literature. Our work uses the same notion that is defined in the textbook _Online computation and competitive analysis_ of Borodin and El-Yaniv and in the seminal paper of Ben-David, Borodin, Karp, Tardos, and Wigderson “On the Power of Randomization in On-Line Algorithms”.  Further, in this case, we cannot hope to get a high probability guarantee better than O(k): consider the case when $f^\star(t) = t/T$ and $f_i(t) = t/T$ when $t < T/k$ and $T/k$ when $t \ge T/k$. Then, since with probability $1/k$, $f^\star(t)$ is the last arm, and so the algorithm can only get $T/k$ reward. Thus, we cannot hope to get better than that reward with arbitrarily high probability.
>
> 2.	The high-level lower bound idea (initial linear increase followed by flattening for most arms, linear increase throughout for optimal arm) is similar between the two works but the actual construction, as well as what is to be shown is completely different. Please see our response to Reviewer WTKj07 for a more thorough discussion of the relationship to their lower bound, but importantly, note that their lower bound for deterministic algorithms only shows that for each algorithm, there exists an instance on which it does poorly, whereas our lower bound for randomized algorithms shows that there exists a distribution over instances on which every algorithm must perform poorly.  Moreover, randomizing over their construction would only yield a constant-factor lower bound for randomized algorithms (see our response to Reviewer WTKj07 for details). Thus, the goals and techniques are distinct. The analysis techniques are also quite different.
>
> We can further isolate assumptions and certain facts in the writeup; thanks for the suggestions in 3, 4, and 5, and thanks for the editorial comments.

---

> > ### Comment · Reviewer_maSC · 2024-11-25
> > **Reply**
> >
> > Thanks for the response.  I'll discuss with the other reviewers and give the final recommendation accordingly.  In the meantime I'll mention the following:
> > - On #1, I still think it's essential to formally state a mathematical equation showing your performance measure, and to immediately discuss its subtleties that make it different from "with high probability" (e.g., a "good" algorithm might still "fail" with high probability)
> > - On #2, perhaps before giving the proof you can give a couple of sentences hinting at what's similar/different to that work

---

### Official Review · Reviewer_WTKj · 2024-11-07
**In favor of acceptance with some reservations**

**Rating:** 6
**Confidence:** 4

**Review:**

## Summary

The paper studies the Improving Multi-Armed Bandits Problem where there are $k$ arms, and each arm's reward is a function of how many times it has been pulled. The authors study this problem when the reward of every arm is increasing and concave in the number of times it has been pulled.

The first result of the paper is that no randomized algorithm can get a competitive ratio better than $\sqrt k /3$. The next result is an algorithm that, given some (approximate) information about the optimal arm, achieves competitive ratio $O(\sqrt k)$. The final main result is another algorithm that runs without the aforementioned information and achieves competitive ratio $O(\sqrt k \log k)$. All of these guarantees are in expectation.


## Class with previous work

The authors claim that Patil et al. (2023) have $\Theta(k)$ upper and lower bounds for the competitive ratio of deterministic algorithms. However, looking at Patil et al. (2023), specifically the proof of Theorem 2 in Appendix C.1 in page 16 (in the arxiv version at least), they seem to be claiming that their $\Omega(k)$ lower bound holds even for randomized algorithms. As far as I can tell, both from skimming the proof of Patil et al. (2023) and the proofs of this paper, Patil et al. (2023) are either wrong or they are considering an adaptive adversary that can respond to the algorithm's actions (which is quite unrealistic). I don't know if the authors of this paper were aware of this technicality (I think they might have been because their lower bound proof is very similar to the one in Patil et al. (2023)), but I'd like to see some discussion about this in the rebuttal.


## Pros

Aside from the above problem, I like the paper. The results are natural since they study the IMAB problem when the algorithms are randomized, which is usually the approach in online algorithms, and especially MAB where Randomization is necessary in some cases to achieve non-trivial bounds. The algorithms are simple and concise and the proofs are simple to understand but not trivial. I found the idea of the $V()$ function quite smart.


## Cons

I was a little puzzled by the final form of Theorem 8. In the last part of the section, the authors change the definition of $m$ so that it holds $m = \Theta(f^*(T-k))$ and $m \le f^*(T-k)$. While this feels similar to the original definition if $k$ is a constant wrt $T$, if $k$ is proportional to $T$, this might lead to quite a big disparity.
Finally, while the authors always made sure to include text that explains both the statements and the proof, in Theorem 8, there is no such text whatsoever. Maybe some explanation for the change in $m$ would have helped with my previous comment.


## Summary

Overall, I liked the paper. I am in favor of accepting, bar the class with previous work which I would like to see discussed in the rebuttal.


## Minor Suggestions

I have the following minor suggestions to the authors:
* The way $m$ is defined initially is confusing. Initially (at the start of Section 4) the authors define $m = \Theta(f^*(T))$ but eventually they require that $m \le f^*(T)$. While this is a small disparity in the text, I think it should have been specified.
* Claim 5: I would suggest rewording to emphasize that the reward comes from that arm: e.g., "the algorithm receives total reward *from that arm* at least..."
* I found the calculations of the minimum in equation (4) a bit tedious. It would have been better if the authors mentioned that it is a second-degree polynomial and that its minimum is attained at point X with value Y.
* The authors seem to be using repeatedly a property of the reward functions (the one mentioned after Equation (12)). If I am right and this property is being used multiple times, it might be useful to make that a claim.

**Paper Award:**

No

---

> ### Author Response · Authors · 2024-11-25
>
> Clash with previous work: their deterministic lower bound does not translate to the randomized setting. For a deterministic lower bound, it is enough to show that for each algorithm, there exists a bad instance. But the bad instance can depend on the algorithm. For a lower bound for randomized algorithms, we must show a distribution over instances that is bad for all algorithms. Their family of hard instances indeed is such that for each deterministic algorithm, there is an instance in this family on which the algorithm will perform poorly. However, the algorithm would actually do fine on a constant fraction of the instances, and so gives only a constant-factor lower bound on the competitive ratio for randomized algorithms..
>
> Theorem 8 / definition of parameter m: this comes down to a subtlety in the way we have defined the recurrence. Note that $V(T’, k’)$ is the expected reward after running the algorithm for $T’ + k’$ steps. This is because if the algorithm plays an arm for $t$ steps, we can only guarantee sufficient reward from $t-1$ of those steps (it takes the $t^\text{th}$ step to figure out this arm is no longer good). Since there are at most $k’$ such extra “switching” steps, the recurrence studies what happens when we give the algorithm those steps for free. Then, in order to translate back to what we actually want to study, we have to compute the outcome of the recurrence with $T’ = T – k$. Thus, the thing we compare to is $f^\star(T-k)$. This point becomes more relevant in the section where we estimate the best value. There, since we spend half the time exploring, we in fact estimate $f^\star(\frac{T}{2} – k)$ and then link it to OPT and to ALG (see proof of Theorem 11). We will add some writing near Theorem 8 to clarify this.
>
> Thank you for your suggestions under “Minor Suggestions” – we will incorporate these into the text.

---

### Official Review · Reviewer_S78y · 2024-11-08
**Review for Submission82**

**Rating:** 8
**Confidence:** 2

**Review:**

*Summary:* This paper studied the improving multi-armed bandit problem (IMAB), where the reward function for each arm is a concave and increasing function with respect to the number of times the arm has been pulled. The goal of the learner is to maximize the cumulative reward given a total number of rounds. This paper provides the first nearly-tight approximation bound for IMAB with randomized algorithm.

*Strength*

1. This paper provides new $\Omega(\sqrt{k})$ lower bound upper bound for an existing problem IMAB, which is an important contribution to this setting.

2. The proposed algorithm matches the bound with mild assumptions. The authors further show how to remove the assumptions at the cost of an $O(\log k)$ factor. This result is significant and achieves better approximation rate compared to the previous work.

*Weakness*

It would be better if the authors could provide more motivation of the problem studied in this paper. Also, the paper could be more comprehensive if the stochastic-reward setting is discussed.

**Paper Award:**

No

---

> ### Author Response · Authors · 2024-11-25
>
> Thank you for your favorable review and feedback. We will add more discussion of motivation to the intro (which so far covers several motivating examples). In terms of the stochastic reward setting, we do have a result for noisy rewards in the appendix.

---

### Meta-Review · Area_Chair_d54R · 2024-12-12

**Recommendation:** Accept
**Confidence:** 4

**Metareview:**

The paper makes interesting advances in the study of the improving multi-armed bandit problem. It highlights a distinction between the performance of deterministic and randomized algorithms. The authors prove new lower bounds for randomized algorithms, and introduce methods that (almost) match the lower bound.
The results are novel and interesting and the paper is a good fit for ALT. On the other hand, the clarity of parts of the paper should be improved for the final version.
In particular, as remarked by a reviewer, the authors should define the performance metric used by their algorithm more clearly and they should discuss more the difference between high probability and expectation guarantees.

**Paper Award:**

No